# Factors Associated with Job Stress and Their Effects on Mental Health among Nurses in COVID-19 Wards in Four Hospitals in Korea

**DOI:** 10.3390/healthcare11101500

**Published:** 2023-05-22

**Authors:** Insu Kim, Hae Ran Kim

**Affiliations:** Department of Nursing, Graduate School of Chosun University, Gwangju 61452, Republic of Korea; revivali@naver.com

**Keywords:** COVID-19, nurse, job stress, anxiety, depression

## Abstract

Increased workload during the COVID-19 pandemic has threatened nurses’ mental health. This study aimed to identify factors associated with job stress in COVID-19 nurses compared to other nurses. Nurses were recruited from four hospitals in Republic of Korea in November 2020. The general sociodemographic questionnaire, job stress, anxiety (GAD-7), and depression (PHQ-9) were used to conduct an online survey. Stepwise multiple regression analysis was used to identify the factors associated with job stress. A total of 290 participants were analyzed: 122 in the dedicated ward and 168 in the nondedicated ward nurse groups. Job stress, anxiety, and depression were higher in nurses dedicated to COVID-19 (4.19 ± 0.59, 5.98 ± 3.92, and 6.97 ± 4.47, respectively) than in the nondedicated group (3.92 ± 0.72 (*p* = 0.001), 4.98 ± 4.20 (*p* = 0.042), and 5.92 ± 4.36 (*p* = 0.047), respectively). Among COVID-19 nurses, job stress levels were higher in 30–39 year olds than in 20–29 year olds (3.71 ± 0.43 vs. 4.04 ± 0.54, *p* = 0.006) and in non-smokers compared with smokers (3.85 ± 0.49 vs. 3.38 ± 0.53, *p* = 0.24). Anxiety (β = 0.34, standard error (SE) = 0.01, *p* < 0.001) and clinical experience of 5–10 years (β = 0.23, SE = 0.10, *p* = 0.004) were associated with job stress. These findings can be applied when devising response strategies for infectious diseases and developing psychological and organizational intervention programs for alleviating job stress in nurses.

## 1. Introduction

The coronavirus disease 2019 (COVID-19) that emerged in 2019 in Wuhan, China, is caused by severe acute respiratory syndrome coronavirus 2 (SARS-CoV-2) [1]. The main route of COVID-19 infection is respiratory droplets through coughing, sneezing, and physical contact [2]. In Republic of Korea, the response to COVID-19 involved the Central Disaster and Safety Countermeasure Headquarters with a focus on disinfection measures [3], and 80% of the Korean population was given the second vaccination [4]. Despite these measures, the number of confirmed patients and deaths have steadily increased due to the continuous spread of COVID-19. As of 14 October 2022, the cumulative numbers were 25,076,239 confirmed cases and 28,783 deaths in Republic of Korea [4].

COVID-19 led to financial and economic losses worldwide and has become a critical disease in public healthcare [5]. The rapid increase in the number of patients and a high rate of mortality resulted in unsafe and inefficient healthcare services [6]. Nurses are the predominant healthcare workers in contact with patients and they have served a crucial role amongst medical professionals in the disaster situation caused by COVID-19 [7]. Among the healthcare professionals committed to the adaptation to and implementation of new protocols related to COVID-19 and the prevention, isolation, management, monitoring, and identification of exposed persons, nurses account for the highest proportion due to the rapid increase in patients requiring care [8]. As nurses devoted themselves to their roles, COVID-19 posed unprecedented threats to the healthcare system to induce various burdens on nurses. Since the outbreak of COVID-19, nurses have been expected to take more care during the use of medical devices, hand hygiene, medical waste disposal, sterilization of treatment equipment, and management of exposed persons [9].

Notably, increased job demands have induced stress at varying levels across hospital nurses. Job stress is defined as the stress induced by excessive job demands that overpower personal resources [10]. Job stress in nurses refers to a state of physiological and psychological damage caused by the excessive demands of the job [11], which can negatively affect the overall outcomes of patient safety and nursing quality [12]. During the COVID-19 pandemic, job stress in hospital nurses was influenced by individual professional competence and various factors of personal and organizational levels [13]. According to a study conducted before the COVID-19 pandemic, age, highest level of education, career length, position, and anticipated role within the department are the known factors associated with job stress [14,15]. During the COVID-19 pandemic, response style and mental health have been identified as the factors associated with job stress [16].

Nurses have the highest risk of COVID-19 exposure and mortality among healthcare workers and they fall in the high-risk group with potential negative mental health results [17,18]. The anxiety that they may infect others, the fear that they may be infected by the patients, and the increased mortality of patients under their care are factors that deteriorate the mental health of dedicated nurses for COVID-19 patients [19]. When COVID-19 was prevalent in Republic of Korea, new nurses reported stress due to the fear of infection for themselves or their families, as well as physical and psychological burdens [20]. The mental health problems of these nurses could also increase due to insufficient staffing at work, inadequate supply of medical equipment, and unsafe working environments [21]. According to the Korean Nurses’ Health Study conducted in 2020, an association was reported between caring for COVID-19 patients and experiencing fear, anxiety, and depression symptoms [22]. The increased mental health problems in nurses and workload due to COVID-19 have been reported as important factors influencing job stress [23]. According to a recent study, the anxiety and depression in frontline nurses exposed to COVID-19 could negatively affect their work performance and potentially lead to increased job stress [16].

Infectious respiratory diseases, such as COVID-19, bring about various changes in the nurses’ roles, practices, and patient care, based on which they have been reported to exert a substantial influence on the turnover or retirement of nurses [24,25]. Securing required resources for the care of patients with new infectious diseases and maintaining manpower is a key strategy in the response to healthcare crises. Thus, it is important to understand the job stress and mental health of nurses caring for COVID-19 patients. The management of job stress is necessary to maximize the work efficiency at hospitals against infectious diseases and advance the nursing organization. In this context, many studies have been conducted to identify the factors associated with job stress in hospital nurses during the COVID-19 pandemic. These studies have a wide spectrum, and a greater level of empirical findings should be achieved to develop strategies applicable to hospital conditions.

Thus, this study compared the level of job stress and mental health problems between dedicated nurses for COVID-19 patients and nurses caring for non-COVID-19 patients in Republic of Korea and investigated the factors that influence job stress in dedicated nurses for COVID-19 patients. The purpose of this study was to provide basic data for developing strategies to manage the job stress in hospital nurses responsible for the care of COVID-19 patients upon an outbreak.

## 2. Methods

### 2.1. Study Design and Participants 

This cross-sectional study was performed in line with the principles of the Declaration of Helsinki. This study obtained approval from the Chosun University Institutional Review Board in Republic of Korea. To conduct the survey, the purpose of this study was explained to the head of the four hospitals. Survey collection took place in November 2020, and Google Forms software was used to manage the questionnaires. The survey link was randomly distributed to all nurses in the four hospitals. Participants were instructed to fill out a self-report questionnaire after obtaining informed consent. The researchers of this study used Google Forms to collect the completed questionnaires, which included the participants’ online consent. The participants who completed the questionnaire were given a token of appreciation (online coupon).

G*Power 3.1.9 program was used to calculate the sample size. The minimum sample size was calculated to be 255 people with the following conditions. Accounting for a dropout rate of 20%, 306 questionnaires were distributed. Of these, 94.8% were collected and 290 questionnaires were analyzed (dedicated nurses for COVID-19 patient: 122, nondedicated: 168).

### 2.2. Instrumentation

The study employed a survey questionnaire adopted from various scales and consisted of 63 items. The general characteristics included sex (male or female), age (20–29, 30–39, or ≥40), marital status (single or married), having children (yes or no), highest level of education (college, university or higher), religion (yes or no), total clinical experience (<1, 1 to <5, 5 to <10, 10 to <15, 15 to <20, or ≥20), department (general ward or special unit), position (staff nurse, charge or head nurse), frequency of drinking (none, 2–3/month, 1–2/week, or ≥3–4/week), current smoking (never, current or former), and frequency of exercise (none, 1–2/week, or ≥3–4/week). Variables were classified according to the definition in the Korean Working Conditions Survey [26]. 

Job stress was measured with a tool developed by Choi (2009) [27]. This tool consisted of work overload (7 items), role conflict as a profession (3 items), lack of professional knowledge and skill (4 items), psychological burden due to the limitation of medicine (5 items), interpersonal relationship (7 items), inadequate compensation (6 items), and physical environment of unit (3 items). The items are on a 5-point Likert scale, and the total score ranges from 1 to 5. Higher scores indicate higher job stress. The tool’s reliability was demonstrated by a Cronbach’s α of 0.95 in this study. 

Mental health problems included anxiety and depression. Anxiety was measured with a Generalized Anxiety Disorder-7 (GAD-7) developed by Spitzer et al. (2006) [28]. Items are on a 4-point Likert scale, and the total score ranges from 0 to 21. A higher score means a higher degree of anxiety and is classified as minimal (0–4 points), mild (5–9 points), moderate (10–14 points), and severe (15–21 points). The tool’s reliability was demonstrated by a Cronbach’s α of 0.90 in this study. Depression was measured with a Patient Health Questionarie-9 (PHQ-9) developed by Kroenke et al. (2001) [29]. Items are on a 4-point Likert scale, and the total score ranges from 0 to 27. A higher score means a higher degree of depressive symptoms and is classified as minimal (0–4 points), mild (5–9 points), moderate (10–14 points), moderately severe (15–19 points), and severe (20–27 points). The tool’s reliability was demonstrated by a Cronbach’s α of 0.86 in this study.

### 2.3. Statistical Data Analysis

All statistical analyses were conducted using SPSS Version 27 (IBM Corp, Armonk, NY, USA). The χ^2^ test and Fisher’s exact test were used to investigate the differences in general characteristics, job stress, and mental health problems between dedicated nurses for COVID-19 patients and the nondedicated groups. Among dedicated nurses for COVID-19 patients, the χ^2^ test and Fisher’s exact test were used to investigate the difference in job stress according to general characteristics. Pearson’s correlation coefficient was used to investigate the correlation between job stress and mental health problems among dedicated nurses for COVID-19 patients. Stepwise multiple regression analysis was used to identify factors affecting job stress in dedicated nurses for COVID-19 patients. Only items with *p* < 0.05 in the univariate analysis were input into the regression analysis. 

## 3. Results

Of dedicated nurses for COVID-19 patients, 77.6% had an education level of university or higher, and this result was higher than that of the nondedicated group (73.25%) (*p* = 0.036). The proportion of nurses with less than 5 years of clinical experience was 62.3% among dedicated nurses for COVID-19 patients and 51.2% in the nondedicated group (*p* = 0.026). The proportion of staff nurses was 88.5% in the COVID-19-dedicated nurse group and 70.2% in the nondedicated group (Table 1). 

The work overload of dedicated nurses for COVID-19 patients (4.19 ± 0.59) was significantly higher than that of the nondedicated group (3.92 ± 0.72) (*p* = 0.001). The anxiety (*p* = 0.042) and depression (*p* = 0.047) scores of COVID-19-dedicated nurses were significantly higher than those of the nondedicated group (Table 2). 

Among dedicated nurses for COVID-19 patients, job stress was significantly higher at 30–39 years of age (4.04 ± 0.54) than at 20–29 years (3.71 ± 0.43) (*p* = 0.006). Job stress in current or former smokers was significantly lower than in the nonsmoker group (*p* = 0.024) (Table 3).

The correlation between job stress, anxiety, and depression among dedicated nurses for COVID-19 patients was positive (Table 4). The analysis of the influence of independent variables on job stress showed that the predictive factors were anxiety (β = 0.34, *p* < 0.001) and total clinical experience (5 to <10 years) (β = 0.23, *p* = 0.004). The coefficient of determination (R^2^) was 0.23, indicating the explanatory power of this model was 23.0%, and the regression model was statistically significant (F = 13.26, *p* < 0.001) (Table 5). To identify the factors affecting the job stress of dedicated nurses for COVID-19 patients, a stepwise multiple regression analysis was performed considering significant variables, such as anxiety, depression, age, total clinical experience, and current smoking status. Nominal measures, such as age, total clinical experience, and current smoking status, were converted into dummy variables. Using dummy variables, nominal variables can be included in the regression model as independent variables. The regression model will estimate the effect of each level of the nominal variable on the dependent variable, relative to the reference level. The analysis of the correlation between the independent variables showed that the correlation coefficient was less than 0.80, confirming that they were independent of each other, and all variables were added to the analysis. When diagnosing a case, there were no outliers greater than the absolute value of 3, and 122 patients were analyzed using the stepwise selection method. The basic assumption test of linear regression analysis was multicollinearity diagnosis, residuals, and outliers. As a result of the analysis, there was no autocorrelation because the Durbin-Watson statistic was 2.11. The tolerance limit was 0.84 to 0.99, which was more than 0.1, and the variance expansion index was 1.00 to 1.20. The independent and dependent variables were normally distributed and the equal variance of individual residuals was also satisfied based on the normal P–P curve and the residual scatterplot. Cook’s distance value did not appear to exceed 1.0 and the regression analysis result was confirmed to be valid.

## 4. Discussion

In this study, the levels of job stress and mental health problems were compared between dedicated nurses for COVID-19 patients and other nurses to identify the factors that influence the job stress in dedicated nurses for COVID-19 patients.

The mean score for job stress was 3.83 for dedicated nurses for COVID-19 patients and 3.92 for general ward nurses, which were higher than the score (3.48) of hospital nurses in Republic of Korea in 2019 before the COVID-19 pandemic [30]. Among the subcategories of job stress, work overload showed a higher score in the group of dedicated nurses for COVID-19 patients than in the control group. It is already known that the workload has increased since the COVID-19 pandemic, which increased job stress in most nurses working at hospitals [11,20]. As COVID-19 is highly contagious, the consequent exponential increase in COVID-19 patients has led to higher levels of physical burden and work intensity due to the lack of healthcare staff, working overtime, and wearing personal protective equipment, which gave rise to work overload in nurses at dedicated COVID-19 patient wards. These working conditions have the potential to increase job stress in these nurses [31]; thus, it was necessary to develop a manual of response and balanced assignment of the nursing manpower caring for infected patients to reduce the workload in nurses. 

The mean scores for anxiety and depression were higher in the group of dedicated nurses for COVID-19 patients than in the control group. In a previous study conducted in Republic of Korea with the same analysis tools, anxiety and depression scores were 2.2 and 4.6 at respective higher levels [32]. This could be attributed to the difference in the time of data collection. The data in the previous study [32] were collected in April 2020, whereas the data in this study were collected in November 2020, when the number of COVID-19 cases rapidly increased [33]. An increase in patients with infection or potential risk of infection consequently affects the workload for nurses [21,34], and this sudden increase in workload can negatively affect their mental health. Alongside the frequent exposure to infected patients, it can also further exacerbate the fear and anxiety in nurses towards infection and death [19]. Additionally, the scores of anxiety and depression in the nondedicated group, who do not have face-to-face contact with the infected patients, were higher than those in a previous study [32]. Irrespective of being dedicated to COVID-19 patients, the nurses periodically tested for COVID-19 while facing an increased workload, a high level of preventive measures, and fear of infection [35], which negatively affects their mental health.

Among the dedicated nurses for COVID-19 patients examined in this study, 57.4% experienced a mild or higher level of anxiety and the percentage of those with a moderate to severe level of anxiety was higher (15.6%) than the control group (10.2%). The level of depression experienced by these nurses was mild for 65.6% and moderate to severe for 25.4%, a level higher than that shown by the control group (16.1%). With the increased risk of COVID-19 exposure in healthcare professionals, the rate of COVID-19 infection was known to be the highest in nurses among healthcare workers [36]. In a variety of studies, dedicated nurses for COVID-19 patients demonstrated a high prevalence of mental health problems, as they were continuously exposed to high workloads accompanied by high rates of infection and death [25,31,32,34]. The characteristic life-threatening side of the disease and increased workload could become a risk factor for high levels of anxiety and depression in dedicated nurses for COVID-19 patients [37]. Meanwhile, positive support from the government and society could protect and improve the mental health of these nurses [38].

In line with previous studies [11,39], the level of job stress was higher in nurses aged 30–39 years than in those aged 20–29 years among dedicated nurses for COVID-19 patients. In Republic of Korea, clinical nurses aged 30–39 years are staff nurse leaders, which entails the highest workload [40]. Through an emergency situation, such as COVID-19, a far greater level of nursing performance was expected of these nurses with a higher proficiency in patient care than new nurses and, as a result, work overload could have increased the rate of burnout and job stress [39]. Hence, adjustments should be made in the administration and structure of hospital nurse organizations, from balanced work allocation to reward systems that can lower the burden of position or work in nurses.

The level of job stress was higher in nonsmoker nurses than in those who formerly or currently smoked. This coincided with a previous study conducted in Republic of Korea [41]. Job stress was caused by contact with infected patients, fatigue, and lack of physical and psychological rest, and smoking could have been chosen as a coping strategy against such stress [42]. For instance, a smoker might have resorted to smoking as a means to instantly relieve stress rather than acting toward improved health. For an individual in a stressful situation, a negative measure such as smoking is known to exert a negative effect on mental health [43]. A positive measure would have the nurses actively seek help from family or society, which could reduce the job stress caused by the COVID-19 pandemic [44].

The factors associated with job stress in dedicated nurses for COVID-19 patients were identified as anxiety and clinical experience of 5–10 years. According to a previous study, negative mental health states, such as anxiety, in a pandemic could reduce the nurses’ resilience, and reduced resilience could be a factor of job stress [7]. As previously mentioned, the data in this study were collected in November 2020, when COVID-19 was spreading rapidly [33]. This could have been a factor of severe anxiety in nurses [19] with a level of impact on job stress. To reduce job stress in nurses, anxiety in nurses should be decreased through reliable information support, adequate supply of personal protective equipment, and rest [45].

With the continuous stream of COVID-19 patients, nurses were required to take an increased number of assigned patients, hygienic procedures, respiratory care, infection prevention care, and frequent overtime work [6]. Compared to new or staff nurses, those with 5–10 years of clinical experience, adequate patient experience, and work efficiency are more likely to be assigned to the wards dedicated to COVID-19 patients [46]. As such, the work-family imbalance, risk of family infection, and health threats met by the nurses with 5–10 years of clinical experience, who have been assigned to the wards dedicated to COVID-19 patients, could be another factor of job stress [47]. Job stress in nurses could lead to turnover intention [48]. The dedicated nurses for COVID-19 patients with 5–10 years of clinical experience have, in fact, exhibited the highest turnover rate compared with other nurses in the same age range [25]. In response to infectious diseases, such as COVID-19, the work conditions of the nurses assigned to the wards dedicated to infected patients should be improved, from work intensity to welfare and livelihood, to reduce the job stress caused by infectious diseases.

This study was able to compare the job stress and mental health problems in dedicated nurses for COVID-19 patients against other nurses during the rapid spread of COVID-19 in Republic of Korea and identify the factors associated with job stress in these nurses. These results will prove valuable in clinical practice and studies towards the development of job stress intervention programs and mental health management strategies for nurses during the pandemic. Thus, this study proposes that a more definitive theoretical model should be developed regarding the factors associated with job stress in nurses dedicated to caring for patients with infectious diseases.

This cross-sectional study has several limitations. First, it was a cross-sectional investigation, and the cause-effect relations could not be accurately defined. Second, due to the small sample size, the number of predictors to be included in the regression model could have been limited as a result of general variables. Third, due to the sample size and the study design, the results can only be extrapolated to the population of nurses in the four institutions where the study was conducted. Our results need to be further explored in larger cohorts. Finally, according to a study conducted during the outbreak of COVID-19, job stress in nurses could be influenced by the nurses’ resilience, self-efficacy, job control and instability, interpersonal conflict, burnout, and turnover intention [49]. However, the variables in this study did not include these factors. Therefore, further studies should be conducted to examine different confounders related to job stress, including these factors.

## 5. Conclusions

The results of this study showed that dedicated nurses for COVID-19 patients could perceive a higher level of job stress with a potentially negative effect on the nurses’ quality of life and patient care, as well as exhibiting higher levels of anxiety and depression. The factors associated with job stress were anxiety and clinical experience of 5–10 years. Job stress threatens the physical and mental health of nurses, reduces their work productivity, and could ultimately exert a negative effect on patient outcomes.

In light of the findings, the following research questions can be explored in future studies: (1) How does the level of job stress and mental health problems among dedicated nurses for COVID-19 patients compare across different countries or healthcare settings? (2) What are the specific organizational and individual factors that contribute to job stress in nurses during infectious disease outbreaks? (3) How can interventions and support programs be developed to mitigate job stress and improve the mental well-being of nurses in high-stress environments, such as COVID-19 wards?

## Figures and Tables

**Table 1 healthcare-11-01500-t001:** General characteristics between dedicated nurse for COVID-19 patients and nondedicated subjects.

Characteristics	Total (n = 290)	Dedicated Nurse for COVID-19 Patients (N = 122)	Nondedicated Nurses (N = 168)	χ^2^	*p*
Sex					
Male	22 (7.6)	13 (10.7)	9 (5.4)	2.83	0.093
Female	268 (92.4)	109 (89.3)	159 (94.6)		
Age (years)					
20–29	173 (59.7)	73 (59.8)	100 (59.5)	2.13	0.346
30–39	78 (26.9)	29 (23.8)	49 (29.2)		
40 ≤	39 (13.4)	20 (16.4)	19 (11.3)		
Marital status					
Single	200 (69.0)	86 (70.5)	114 (67.9)	0.23	0.632
Married	90 (31.0)	36 (29.5)	54 (32.1)		
Having children					
No	214 (73.8)	87 (71.3)	127 (75.6)		
Yes					
Highest level of education					
College	65 (22.4)	20 (16.4)	45 (26.8)	4.39	0.036
≥University	225 (77.6)	102 (83.6)	123 (73.2)		
Religion					
No	215 (74.1)	87 (71.3)	128 (76.2)	0.88	0.349
Yes	75 (25.9)	35 (28.7)	40 (23.8)		
Total clinical experience (year)					
<1	37 (12.8)	11 (9.0)	26 (15.5)	11.08	0.026
1 - < 5	125 (43.1)	65 (53.3)	60 (35.7)		
5 - < 10	60 (20.7)	24 (19.7)	36 (21.4)		
10 - < 15	34 (11.7)	9 (7.4)	25 (14.9)		
15 ≤	34(11.7)	13(10.6)	21(12.5)		
Department					
General ward	179 (61.7)	69 (56.6)	110 (65.5)	2.38	0.123
Special unit	111 (38.3)	53 (43.4)	58 (34.5)		
Position					
Staff nurse	226 (77.9)	108 (88.5)	118 (70.2)	13.74	< 0.001
Charge and Head nurse	64 (22.1)	14 (11.5)	50 (29.8)		
Frequency of drinking					
None	124 (42.8)	52 (42.6)	72 (42.9)	3.43	0.330
2–3/month	100 (34.5)	41 (33.7)	59 (35.1)		
1–2/week	56 (19.3)	22 (18.0)	34 (20.2)		
3–4/week ≤	10 (3.4)	7 (5.7)	3 (1.8)		
Current smoking status					
Never	279 (96.2)	116 (95.1)	163 (97.0)	-	0.536 *
Current and former	11 (3.8)	6 (4.9)	5 (3.0)		
Frequency of exercise					
None	166 (57.2)	71 (58.2)	95 (56.5)	2.80	0.246
1–2/week	98 (33.8)	44 (36.1)	54 (32.1)		
3–4/week ≤	26 (9.0)	7 (5.7)	19 (11.3)		

Data represent n (%) and mean ± standard deviation. * Fisher exact test. COVID-19 = Coronavirus Disease-19.

**Table 2 healthcare-11-01500-t002:** Job stress and mental health problems status between dedicated COVID-19 patients’ nurses and nondedicated subjects.

Characteristics	Total	Dedicated Nurse for COVID-19 Patients	Nondedicated Nurses	χ^2^	*p*
Job stress	3.78 ± 0.58	3.83 ± 0.50	3.74 ± 0.63	1.30	0.196
Work overload	4.03 ± 0.68	4.19 ± 0.59	3.92 ± 0.72	3.48	0.001
Role conflict as a profession	3.46 ± 0.78	3.55 ± 0.69	3.40 ± 0.84	1.64	0.103
Lack of professional knowledge and skill	4.10 ± 0.68	4.07 ± 0.62	4.13 ± 0.72	−0.71	0.478
Psychological burden due to the limitation of medicine	3.64 ± 0.68	3.67 ± 0.65	3.62 ± 0.70	0.61	0.543
Interpersonal relationship	3.71 ± 0.80	3.76 ± 0.73	3.68 ± 0.85	0.81	0.420
Inadequate compensation	3.78 ± 0.72	3.83 ± 0.65	3.75 ± 0.76	0.95	0.344
Physical environment of unit	3.70 ± 0.81	3.75 ± 0.73	3.66 ± 0.86	0.92	0.359
Mental health					
Anxiety	5.40 ± 4.12	5.98 ± 3.92	4.98 ± 4.20	2.04	0.042
Minimal (0–4 point)	138 (4 7.6)	52 (42.6)	86 (51.2)	5.90	0.116
Mild (5–9 point)	116 (40.0)	51 (41.8)	65 (38.7)		
Moderate (10–14 point)	26 (9.0)	16 (13.1)	10 (6.0)		
Severe (15–21 point)	10 (3.4)	3 (2.5)	7 (4.2)		
Depression	6.36 ± 4.43	6.97 ± 4.47	5.92 ± 4.36	1.99	0.047
Minimal (0–4 point)	107 (36.9)	42 (34.4)	65 (38.7)	-	0.390 ^*^
Mild (5–9 point)	125 (43.1)	49 (40.2)	76 (45.2)		
Moderate (10–14 point)	42 (14.5)	23 (18.9)	19 (11.3)		
Moderately severe (15–19 point)	12 (4.1)	6 (4.9)	6 (3.6)		
Severe (20–27 point)	4 (1.4)	2 (1.6)	2 (1.2)		

Data represent n (%) and mean ± standard. * Fisher exact test. COVID-19 = Coronavirus Disease-19.

**Table 3 healthcare-11-01500-t003:** Differences in job stress by general characteristics among dedicated nurse for COVID-19 patients.

Characteristics	N (%)	M ± SD	t or χ^2^	*p* (Post Hoc ^†^)
Sex				
Male	13 (10.7)	3.62 ± 0.53	−1.63	0.105
Female	109 (89.3)	3.86 ± 0.50		
Age (years)				
20–29 ^a^	73 (59.8)	3.71 ± 0.43	5.38	0.006
30–39 ^b^	29 (23.8)	4.04 ± 0.54		(b > a)
40 ≤ ^c^	20 (16.4)	3.95 ± 0.60		
Marital status				
Single	86 (70.5)	3.78 ± 0.47	−1.63	0.107
Married	36 (29.5)	3.94 ± 0.57		
Having children				
No	87 (71.3)	3.79 ± 0.47	−1.28	0.203
Yes	35 (28.7)	3.92 ± 0.57		
Highest level of education				
College	20 (16.4)	3.98 ± 0.49	1.48	0.142
≥University	102 (83.6)	3.80 ± 0.50		
Religion				
No	87 (71.3)	3.82 ± 0.47	−0.26	0.798
Yes	35 (28.7)	3.85 ± 0.60		
Total clinical experience (year)				
<1	11 (9.0)	3.54 ± 0.45	2.93	0.024
1 - < 5	65 (53.3)	3.76 ± 0.48		(none)
5 - < 10	24 (19.7)	4.07 ± 0.46		
10 - < 15	9 (7.4)	4.00 ± 0.72		
15 ≤	13 (10.6)	3.85 ± 0.41		
Department				
General ward	69 (56.6)	3.81 ± 0.52	−0.52	0.605
Special unit	53 (43.4)	3.86 ± 0.49		
Position				
Staff nurse	108 (88.5)	3.82 ± 0.50	−0.80	0.427
Charge and Head nurse	14 (11.5)	3.93 ± 0.50		
Frequency of drinking				
None	52 (42.6)	3.86 ± 0.50	1.34	0.266
2–3/month	41 (33.7)	3.88 ± 0.52		
1–2/week	22 (18.0)	3.75 ± 0.46		
3–4/week ≤	7 (5.7)	3.52 ± 0.52		
Current smoking status				
Never	116 (95.1)	3.85 ± 0.49	2.28	0.024
Current or former	6 (4.9)	3.38 ± 0.53		
Frequency of exercise				
None	71 (58.2)	3.79 ± 0.52	0.61	0.545
1–2/week	44 (36.1)	3.89 ± 0.48		
3–4/week ≤	7 (5.7)	3.87 ± 0.57		

Data represent n (%) and mean ± standard. ^a^, ^b^, ^c^, Post hoc ^†^: Scheffé test.

**Table 4 healthcare-11-01500-t004:** Correlation between job stress and mental health problems among dedicated nurses for COVID-19 patients.

Variables	Job Stress r (*p*)	Anxiety r (*p*)
Anxiety	0.42 (<0.001)	
Depression	0.28 (0.002)	0.70 (<0.001)

**Table 5 healthcare-11-01500-t005:** Factors influencing job stress among dedicated nurses for COVID-19 patients.

Variable	B	SE	β	t	*p*	Adj R^2^	F	*p*
Constant	3.27	0.12		27.75	<0.001	0.23	13.26	<0.001
Anxiety	0.04	0.01	0.34	3.90	<0.001			
Total clinical experience (5 - < 10 years)	0.29	0.10	0.23	2.92	0.004			

Including variables in stepwise method: anxiety and depression, age = dummy variable (20–29 year = 1), total clinical experience = dummy variable (<1 year = 1), current smoking status = dummy variable (Never = 1). SE=standard error; Adj = adjusted.

## Data Availability

The data of the current study are available from the corresponding author upon reasonable request.

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
