# Peer review of "Factors Associated with Job Stress and Their Effects on Mental Health among Nurses in COVID-19 Wards in Four Hospitals in Korea"

_healthcare, 2023, doi:10.3390/healthcare11101500_

Round 1
Reviewer 1 Report
I provide here some comments that might improve the quality of the article.
Regarding the title of the work, I would suggest the following: “Factors associated with job stress, and effects on mental health among nurses in wards dedicated to COVID-19 patients in Korea.” It would better respond to the proposed objective.
It is recommended to place the sample size in the abstract.
The type of training required for the nursing profession in Korea should be briefly explained, since the introduction of the variable "educational level" is confusing when it establishes levels below university as possible ("College" vs. "University").
Regarding the representativeness of the sample calculated using "GPower", even though it is adequate for the total sample, it would be interesting to provide data on the representativeness by gender, given the large difference in the sample size of both.
Briefly explain the reason for using the "dummy variables" and the points chosen for the "dichotomization" of them.
Author Response
Manuscript ID: healthcare-2352928
Type of manuscript: Article
Title: Factors associated with job stress among nurses in wards dedicated to COVID-19 patients in Korea
Authors: Insu Kim, Hae Ran Kim *
`
Thank you for reviewing this study. We have attempted to address the reviewer's comments and improve the manuscript. We have included point-by-point responses to each comment in this resubmission. The detailed responses to each comment are as follows:
Response to Reviewer 1 Comments.
I provide here some comments that might improve the quality of the article.
Regarding the title of the work, I would suggest the following: “Factors associated with job stress, and effects on mental health among nurses in wards dedicated to COVID-19 patients in Korea.” It would better respond to the proposed objective.
Reply:
- I appreciate your suggestion regarding the title of my work. We agree that your proposed title. The title has been changed as follows
Title
Factors associated with job stress, and effects on mental health among nurses in wards dedicated to COVID-19 patients in Korea
It is recommended to place the sample size in the abstract.
Reply:
- We will make the necessary changes to the abstract of my thesis and include the sample size information as recommended. By doing so, we hope to enhance the overall quality and impact of research.
Abstract
A total of 290 participants were analyzed, with 122 in the dedicated ward nurse group and 168 in the non-dedicated ward group.
The type of training required for the nursing profession in Korea should be briefly explained, since the introduction of the variable "educational level" is confusing when it establishes levels below university as possible ("College" vs. "University").
Reply:
- To avoid confusion, it has been changed to 'Highest level of education'. We hope to provide readers with a better understanding of the educational background of the participants in this study and to eliminate any potential sources of confusion.
Regarding the representativeness of the sample calculated using "GPower", even though it is adequate for the total sample, it would be interesting to provide data on the representativeness by gender, given the large difference in the sample size of both.
Reply:
- We agree that providing data on the representativeness of the sample by gender would add value to my study, given the significant difference in sample size between male and female participants. However, given that the proportion of male nurses in Korea is low, this study included nurses of all genders. In future studies, we will conduct the relevant study to include a more detailed analysis of the representativeness of the sample by gender.
Briefly explain the reason for using the "dummy variables" and the points chosen for the "dichotomization" of them.
Reply:
- We will explain that "dummy variables" are a statistical tool used to represent categorical variables in regression analysis, and that they allow us to estimate the effect of different categories on the dependent variable. In accordance with the reviewer's feedback, we have added the following sentences. Table 5 footnotes present the options for dichotomization. Please let us know again if we have misunderstood something.
Methods
By using dummy variables, nominal variables can be included in the regression model as independent variables. The regression model will estimate the effect of each level of the nominal variable on the dependent variable, relative to the reference level.
Table 5 footnote
Including variables in stepwise method: Anxiety and depression, age=dummy variable (20~29 year=1), total clinical experience=dummy variable (<1 year=1), current smoking status=dummy variable (Never=1). SE=standard error; Adj=adjusted.

Reviewer 2 Report
I have read your paper titled "Factors associated with job stress among nurses in wards dedicated to COVID-19 patients in Korea," and I would like to provide some feedback:
- While the overall structure of the introduction is well-organized, there seems to be a lack of research perspectives specifically focused on nursing in Korea. Given the title and the context of your study, it would be beneficial to include more information on the existing literature and efforts related to nursing and job stress in the Korean healthcare setting.
- Incorporating studies and reports related to the challenges faced by Korean nurses during the COVID-19 pandemic will not only strengthen the background of your paper but also provide valuable context for the reader. This will help to establish the relevance of your research and emphasize the need for further investigation into the factors associated with job stress among nurses in Korea.
- To address this gap, I recommend that you review and include relevant sources that discuss the nursing profession in Korea, including the working conditions, job demands, and the overall impact of the COVID-19 pandemic on the nursing workforce. By doing so, you will be able to provide a more comprehensive background for your research, which will ultimately enhance the overall quality and impact of your work.
- The clarity and organization in the methods and results sections allow the reader to easily follow and understand the study's design, data collection, analysis, and outcomes.
- I hope you find this feedback helpful as you continue to revise your paper. I look forward to reading the final version and learning more about the factors associated with job stress among nurses in COVID-19 wards in Korea.
- I am pleased to see that the discussion is well-structured and effectively connects your findings to the broader context of nursing and job stress in the COVID-19 pandemic.However, as you revise and update the introduction based on previous feedback, it is essential to ensure that the discussion remains coherent and aligned with any changes made in the introduction. This will ensure that your paper maintains a logical flow and that the discussion is grounded in the context provided by the updated introduction.
- I hope you find this feedback helpful as you continue to refine your introduction. I look forward to reading the final version of your paper and learning more about the factors contributing to job stress among nurses in COVID-19 wards in Korea
Author Response
Manuscript ID: healthcare-2352928
Type of manuscript: Article
Title: Factors associated with job stress among nurses in wards dedicated to COVID-19 patients in Korea
Authors: Insu Kim, Hae Ran Kim *
Thank you for reviewing this study. We have attempted to address the reviewer's comments and improve the manuscript. We have included point-by-point responses to each comment in this resubmission. The detailed responses to each comment are as follows:
Response to Reviewer 2 Comments.
I have read your paper titled "Factors associated with job stress among nurses in wards dedicated to COVID-19 patients in Korea," and I would like to provide some feedback:
- While the overall structure of the introduction is well-organized, there seems to be a lack of research perspectives specifically focused on nursing in Korea. Given the title and the context of your study, it would be beneficial to include more information on the existing literature and efforts related to nursing and job stress in the Korean healthcare setting.
- Incorporating studies and reports related to the challenges faced by Korean nurses during the COVID-19 pandemic will not only strengthen the background of your paper but also provide valuable context for the reader. This will help to establish the relevance of your research and emphasize the need for further investigation into the factors associated with job stress among nurses in Korea.
- To address this gap, I recommend that you review and include relevant sources that discuss the nursing profession in Korea, including the working conditions, job demands, and the overall impact of the COVID-19 pandemic on the nursing workforce. By doing so, you will be able to provide a more comprehensive background for your research, which will ultimately enhance the overall quality and impact of your work.
- The clarity and organization in the methods and results sections allow the reader to easily follow and understand the study's design, data collection, analysis, and outcomes.
- I hope you find this feedback helpful as you continue to revise your paper. I look forward to reading the final version and learning more about the factors associated with job stress among nurses in COVID-19 wards in Korea.
- I am pleased to see that the discussion is well-structured and effectively connects your findings to the broader context of nursing and job stress in the COVID-19 pandemic.However, as you revise and update the introduction based on previous feedback, it is essential to ensure that the discussion remains coherent and aligned with any changes made in the introduction. This will ensure that your paper maintains a logical flow and that the discussion is grounded in the context provided by the updated introduction.
- I hope you find this feedback helpful as you continue to refine your introduction. I look forward to reading the final version of your paper and learning more about the factors contributing to job stress among nurses in COVID-19 wards in Korea
Reply:
- We completely agree with your first comment and recognize the importance of including more research perspectives specifically focused on nursing in Korea. We will review and include the overall impact of the COVID-19 pandemic on the nursing workforce. This will provide a more comprehensive background for our research and emphasize the need for further investigation into the factors associated with job stress among nurses in Korea. To improve the quality of this research, we reviewed mostly SCI or SSCI articles. In the latest articles we reviewed, there were few results regarding job stress among Korean nurses during the pandemic period. We have added relevant references based on the reviewer's feedback and made efforts to integrate them with our topic. In the discussion section, relevant content has been presented.
Introduction
Nurses have the highest risk of COVID-19 exposure and mortality across healthcare workers, and they fall in the high-risk group with potential negative mental health results [17,18]. The anxiety that they may infect others, the fear that they may be infected by the patients, and the increased mortality of patients under their care are the factors that deteriorate the mental health of dedicated nurses for COVID-19 patients [19]. When COVID-19 was prevalent in Korea, new nurses reported stress due to fear of infection for themselves or their families and physical and psychological burdens [20]. The mental health problems of these nurses could also increase due to the insufficient manpower at work, inadequate supply of medical equipment, and unsafe working environments [21]. According to the Korean Nurses' Health Study conducted in 2020, the association was reported be-tween caring for COVID-19 patients and experiencing fear, anxiety, and depression symptoms [22]. The increased mental health problems in nurses as well as workload due to COVID-19 have been reported as important factors influencing job stress [23]. According to a recent study, the anxiety and depression in frontline nurses ex-posed to COVID-19 could negatively affect their work performance and potentially lead to increased job stress [16].
Discussion
The mean score for job stress was 3.83 for dedicated nurses for COVID-19 patients and 3.92 for general ward nurses. The scores were higher than 3.48 shown by hospital nurses in South Korea as measured using the same tool in a study conducted in 2019 be-fore the COVID-19 pandemic [30]. Among the subcategories of job stress, work overload showed a higher score in the group of dedicated nurses for COVID-19 patients than in the control group. It is already known that the workload has increased since the COVID-19 pandemic, which has increased the job stress in most nurses working at hospitals [11,20]. As COVID-19 is highly contagious, the consequent exponential increase in COVID-19 patients has led to higher levels of physical burden and work intensity due to the lack of healthcare staff, working overtime, and wearing personal protective equipment, which in turn gave rise to work overload in nurses working at wards dedicated to COVID-19 patients. This has the potential to increase the job stress in these nurses [31] and it is thus necessary to develop a manual of response and balanced assignment of the nursing manpower caring for infected patients, to reduce the workload in nurses.
The mean scores for anxiety and depression were higher in the group of dedicated nurses for COVID-19 patients than in the control group. In a previous study conducted in South Korea with the same analysis tools, the scores for anxiety and depression were 2.2 and 4.6 at respective higher levels [32]. This could be attributed to the difference in the time of data collection. The data in the previous study [32] were collected in April 2020, where-as the data in this study were collected in November 2020, when the number of COVID-19 cases rapidly increased [33]. An increase in patients with infection or potential risk of infection consequently affects the workload for nurses [21,34] and a sudden in-crease in work-load can have a negative effect on the mental health of nurses. Alongside the frequent exposure to infected patients, it can also further extend the fear and anxiety in nurses toward infection and death [19]. Additionally, the scores of anxiety and depression in the non-dedicated group, who do not have face-to-face contact with the infected patients, were higher than those in a previous study [32]. Irrespective of whether being dedicated to COVID-19 patients, the nurses periodically tested for COVID-19, while facing increased workload, a high level of preventive measures, and fear of infection [35], which could have a negative effect on their mental health.

Reviewer 3 Report
In this study, researchers aimed to determine the factors that contribute to job stress in nurses during the COVID-19 pandemic and compare the stress levels of COVID-19 nurses with those of other nurses. They recruited nurses from four hospitals in South Korea in November 2020 and conducted an online survey that included questions about sociodemographic information, job stress, anxiety, and depression. The results showed that COVID-19 nurses had higher levels of job stress, anxiety, and depression than the control group. Additionally, job stress levels were higher in 30-39-year-olds and non-smokers compared to 20-29-year-olds and smokers, respectively, among COVID-19 nurses. The study also found that anxiety and clinical experience of 5-10 years were associated with job stress. These findings can be used to develop strategies to alleviate job stress in nurses during infectious disease outbreaks and implement psychological and organizational intervention programs.
The researchers in this study used Stepwise multiple regression analysis to identify the factors associated with job stress in COVID-19 nurses. This type of analysis is appropriate when there are multiple predictor variables, and the goal is to identify the most important predictors of a particular outcome. Furthermore, the use of stepwise multiple regression allows for the selection of a subset of predictor variables that are most strongly associated with the outcome, which can be useful for identifying specific factors that contribute to job stress in COVID-19 nurses. Therefore, the choice of Stepwise multiple regression analysis in this study was appropriate given the research question and outcome variable of interest.
While the literature on psychosocial factors associated with healthcare delivery, particularly for COVID-19 frontline workers, has been extensive, it is not entirely clear how much additional value can be gained from further exploration of the topic at this point in the pandemic. While this subject matter may have been of great interest at an earlier stage, the contribution of additional literature on the subject at this point is unclear.
Furthermore, it should be noted that the study did not present a sampling strategy. While the study design was cross-sectional and used terms such as "control" and "exposed," which may suggest a cohort study, this type of research typically requires a different approach to sampling and subsequent analysis. Additionally, the authors did not report whether the survey instrument used in the study underwent internal and external validation. It is also striking that there is no mention of an ethics committee approving the study. These limitations should be taken into consideration when interpreting the findings of this research.
Author Response
Manuscript ID: healthcare-2352928
Type of manuscript: Article
Title: Factors associated with job stress among nurses in wards dedicated to COVID-19 patients in Korea
Authors: Insu Kim, Hae Ran Kim *
Thank you for reviewing this study. We have attempted to address the reviewer's comments and improve the manuscript. We have included point-by-point responses to each comment in this resubmission. The detailed responses to each comment are as follows:
Response to Reviewer 3 Comments.
In this study, researchers aimed to determine the factors that contribute to job stress in nurses during the COVID-19 pandemic and compare the stress levels of COVID-19 nurses with those of other nurses. They recruited nurses from four hospitals in South Korea in November 2020 and conducted an online survey that included questions about sociodemographic information, job stress, anxiety, and depression. The results showed that COVID-19 nurses had higher levels of job stress, anxiety, and depression than the control group. Additionally, job stress levels were higher in 30-39-year-olds and non-smokers compared to 20-29-year-olds and smokers, respectively, among COVID-19 nurses. The study also found that anxiety and clinical experience of 5-10 years were associated with job stress. These findings can be used to develop strategies to alleviate job stress in nurses during infectious disease outbreaks and implement psychological and organizational intervention programs.
The researchers in this study used Stepwise multiple regression analysis to identify the factors associated with job stress in COVID-19 nurses. This type of analysis is appropriate when there are multiple predictor variables, and the goal is to identify the most important predictors of a particular outcome. Furthermore, the use of stepwise multiple regression allows for the selection of a subset of predictor variables that are most strongly associated with the outcome, which can be useful for identifying specific factors that contribute to job stress in COVID-19 nurses. Therefore, the choice of Stepwise multiple regression analysis in this study was appropriate given the research question and outcome variable of interest.
While the literature on psychosocial factors associated with healthcare delivery, particularly for COVID-19 frontline workers, has been extensive, it is not entirely clear how much additional value can be gained from further exploration of the topic at this point in the pandemic. While this subject matter may have been of great interest at an earlier stage, the contribution of additional literature on the subject at this point is unclear.
Furthermore, it should be noted that the study did not present a sampling strategy. While the study design was cross-sectional and used terms such as "control" and "exposed," which may suggest a cohort study, this type of research typically requires a different approach to sampling and subsequent analysis. Additionally, the authors did not report whether the survey instrument used in the study underwent internal and external validation. It is also striking that there is no mention of an ethics committee approving the study. These limitations should be taken into consideration when interpreting the findings of this research.
Reply:
- Our study aimed to investigate the mental health and job stress of dedicated ward nurses during the pandemic. Studies on this topic among Korean nurses are rare. In particular, we found an association between '5 - < 10' clinical experience and job stress in our study. The authors describe this result in the Discussion section.
- We agree with the reviewer's suggestion. It has been changed to 'non-dedicated group' instead of 'control'.
- The general characteristics used in our study followed the categories of the Korean Working Conditions Survey (KWCS). All tools in KWCS are regularly evaluated by experts. Additionally, the tool used for measuring mental health is a traditional tool that has already been validated for its reliability.
- We have added a mention of ethics committee approval.
Round 2
Reviewer 3 Report
In light of the study design and sample size, I would like to respectfully suggest that the title be revised to "Factors associated with job stress and their effects on mental health among nurses in COVID-19 wards in four hospitals in Korea." Additionally, I recommend that the paragraph in the discussion which mentions that "the questionnaire survey in this study could have restricted the subjects to only those with access to an online platform" be revised. It is important to note that due to the size of the sample and the study design, the results can only be extrapolated to the population of nurses in the four institutions where the study was conducted.
Author Response
Manuscript ID: healthcare-2352928
Type of manuscript: Article
Title: Factors associated with job stress among nurses in wards dedicated to COVID-19 patients in Korea
Authors: Insu Kim, Hae Ran Kim *
`
Thank you for reviewing this study. We have attempted to address the reviewer's comments and improve the manuscript. We have included point-by-point responses to each comment in this resubmission. The detailed responses to each comment are as follows:
Response to Reviewer Comments.
In light of the study design and sample size, I would like to respectfully suggest that the title be revised to "Factors associated with job stress and their effects on mental health among nurses in COVID-19 wards in four hospitals in Korea." Additionally, I recommend that the paragraph in the discussion which mentions that "the questionnaire survey in this study could have restricted the subjects to only those with access to an online platform" be revised. It is important to note that due to the size of the sample and the study design, the results can only be extrapolated to the population of nurses in the four institutions where the study was conducted.
Reply:
- We appreciate your suggestion regarding the title of my work. The title has been changed as follows:
Title
Factors associated with job stress and their effects on mental health among nurses in COVID-19 wards in four hospitals in Korea
- As the reviewer pointed out, there would have been almost no restrictions on data collection through online platforms, especially in Korea. Therefore, that sentence has been deleted and the following sentence has been added.
Discussion
Third, since the sample size and the study design, the results can only be extrapolated to the population of nurses in the four institutions where the study was conducted. Our results need to be further explored in larger cohorts.